# First-line risk stratification with machine learning models facilitates rapid triage for non-ST-elevation myocardial infarction

**Wei-Jia Luo[1], Yih-Mei Liou[2], Cheng-Han Hsiao[1], Chi-Sheng Hung[3], Heng-Yu Pan[4], Chien-Hua Huang[5], Pan-Chyr Yang[6,7‡], Kang-Yi Su**[1,2,8]*

**1** Department of Clinical Laboratory Sciences and Medical Biotechnology, College of Medicine, National Taiwan University, Taipei, Taiwan, Republic of China, **2** Department of Laboratory Medicine, National Taiwan University Hospital, Taipei, Taiwan, Republic of China, **3** Division of Cardiology, Department of Internal Medicine, National Taiwan University Hospital, Taipei, Taiwan, Republic of China, **4** Division of Cardiology, Department of Internal Medicine, National Taiwan University Hospital Hsin-Chu Branch, Hsin-Chu, Taiwan, Republic of China, **5** Department of Emergency Medicine, National Taiwan University Hospital, Taipei, Taiwan, Republic of China, **6** Department of Internal Medicine, National Taiwan University Hospital, Taipei, Taiwan, Republic of China, **7** Institute of Biomedical Sciences, Academia Sinica, Taipei, Taiwan, Republic of China, **8** Genome and Systems Biology Degree Program, National Taiwan University and Academia Sinica, Taipei, Taiwan, Republic of China

‡ These authors are joint senior authors on this work.
* suky@ntu.edu.tw

## Abstract

Timely diagnosis of non-ST-elevation myocardial infarction (NSTEMI) remains challenging, as current protocols rely on serial high-sensitivity cardiac troponin (hs-cTn) tests that may delay decisions and overcrowd emergency departments. We retrospectively analyzed 54,636 patients receiving hs-cTn testing at emergency departments across Taiwan (May 2016–Dec 2021). Excluding STEMI and incomplete cases, we developed a machine learning (ML) model using demographics and 23 routine lab tests from the initial blood draw to enable early NSTEMI risk stratification. An actionable clinical decision supporting algorithm was also created based on ML-derived risk scores. A total of 15,096 eligible patients (mean age 69.94 ± 15.66 years; 42.2% female) were included in model training and evaluation. The ML model outperformed hs-cTn alone in both internal and external validation sets in terms of area under the receiver-operating characteristic curve. Beyond model development, a clinically actionable decision algorithm using risk score was established. Thresholds (<1.8 and ≥38.5) to define low- and high-risk groups, the model achieved a negative predictive value (NPV) of 98.8% (98.5–99.1%) for rule-out and a positive predictive value (PPV) of 78.1% (73.2–82.4%) for rule-in, encompassing 48.3% and 2.6% of patients, respectively. When combined with the established 0 h/1 h algorithm, the ML model further enhanced early decision-making, safely ruling in/out 85.3% of patients within 1 hour, with PPV and NPV reaching 84.9% (79.5–87.7%) and 100% (99.6–100%), respectively. In conclusion, this ML-based approach offers not only

**Data availability statement:** Ethical approval was obtained from the Institutional Review Board of National Taiwan University Hospital (NTUH) (Approval No. 202203044RINA). All retrospective clinical data were obtained from the NTUH-Integrative Medical Data Center. Requests for access to the minimal dataset are subject to institutional policies and ethical approval. For non-author contact information regarding data access inquiries, the appropriate institutional body is the NTUH Ethics Center (Phone: +886-2-23123456 ext. 263160; Email: ntuhrec@ntuh.gov.tw; https://www.ntuh.gov.tw/EC/Index.action?l=en_US). Long-term data storage is managed by the NTUH-Integrative Medical Data Center (Phone: +886-2-23123456 ext. 264076; Email: chuangshulin@ntuh.gov.tw; https://www.ntuh.gov.tw/LARD-IMD/Index.action?l=en_US), which securely maintains the data and coordinates responses to external data access requests in accordance with applicable institutional regulations, ethical requirements, and data protection laws.

**Funding:** This work was supported by the National Science and Technology Council, Taiwan (MOST111-2628-B-002-029-MY3 and NSTC114-2320-B-002-066-MY3 to K.Y.S). The funder has no role in the study design, data collection, analysis, interpretation, or writing of the manuscript.

**Competing interests:** The authors have declared that no competing interests exist.

accurate prediction but also an actionable guide to support rapid, safe NSTEMI triage in emergency care.

## Author summary

Every minute counts when someone comes to the emergency room with chest pain, yet identifying a heart attack quickly and accurately remains a major challenge, especially when early test results are inconclusive. In our study, we wanted to improve the initial evaluation of people suspected of having a certain type of heart attack, called non-ST-elevation myocardial infarction, or NSTEMI. We developed computer models that use routine blood tests and basic patient information, collected right when the patient arrives, to predict their risk of having a heart attack. These models worked well, even better than relying on standard blood tests alone, and could identify over half of the patients as either low or high risk within the first blood draw. We also found that combining our model with existing guidelines made early triage faster and more accurate. Our approach could help emergency doctors make quicker decisions, reduce unnecessary waiting, and prioritize care for those who need it most. We believe this kind of machine learning tool, based on real-world data and simple tests, could be a practical step forward in emergency heart care.

## Introduction

Acute myocardial infarction (AMI) refers to cardiomyocyte necrosis after acute myocardial ischemia and can be fatal without immediate medical care [1]. Early diagnosis is crucial to initiate timely treatment and save lives, yet only a small proportion of patients are diagnosed with AMI upon arrival to the emergency department (ED) in the U.S., Europe, and Taiwan [2–4]. High-sensitivity cardiac troponin (hs-cTn) assays provide higher accuracy for AMI diagnosis and allow for more rapid confirmation or exclusion of the condition compared with the traditional cTn tests [5,6]. However, for patients unable to be ruled in/out at the $0^{th}$ hour according to the 0 h/3 h algorithm or ECG, the time interval before the next hs-cTn re-test is approximately 3 hours [7], leading to the delay to diagnosis, prolonged stays, and even congestion in the ED [8]. The European Society of Cardiology (ESC) guideline recommends using the 0 h/1 h algorithm for rapid confirmation/exclusion of patients presenting with suspected non-ST-segment elevation acute coronary syndrome (NSTE-ACS) [9]. Even so, nearly one-third of patients still require observation after the diagnostic cascade [10], highlighting an urgent need for instantaneous and accurate strategies for prompt identification of AMI.

The greatest challenge for AMI diagnosis is the integration of clinical presentation with information on symptoms, ECG, troponin assessment, and imaging modalities into a standardized management strategy [11]. Routine laboratory tests not only provide comprehensive information on physiological conditions but also form the

basis for clinical decision-making. Several hematological and biochemical biomarkers have been associated with mortality in NSTE-ACS evolving into AMI and thus confer an additive diagnostic or prognostic value to cTn [12,13]. Given that no single laboratory test generated from the first blood draw accurately discriminates AMI from non-AMI patients, the combination of these indexes is thus an alternative approach. Recent progress in the development of artificial intelligence-based algorithms as diagnostic tools has made it feasible to use medical data to construct machine learning (ML) models [14–16]. Diagnostically relevant values for various acute and chronic diseases have been determined for combinations of routine laboratory tests, with ML algorithms identifying nonlinear relationships between model features and outcomes [17,18]. More recently, an ML algorithm incorporating age, sex, and serial hs-cTn collected at different time points to predict the likelihood of AMI was introduced and validated [19,20]. However, time-consuming and cost considerations remain as these algorithms still rely on a second blood draw and a second hs-cTn measurement. We aimed to develop ML models integrating demographic factors, first hs-cTn, and routine laboratory test data at the ED for rapid AMI risk assessment.

## Materials and methods

### Ethics statement

This retrospective study complied with the principles outlined in the Declaration of Helsinki and followed the TRIPOD guideline for experimental design. Ethical approval was obtained from the Institutional Review Board (IRB) of NTUH (Approval No. 202203044RINA). Given the use of anonymized data from an authorized biobank, the study was deemed minimal risk, so the IRB granted an exemption from obtaining individual informed consent.

### Study design

We developed the NSTEMI ML model by retrospectively collecting the data from 44,000 patients who had undergone at least one hs-cTnT measurement at the ED of National Taiwan University Hospital (NTUH) from 1 May, 2016 to 31 Dec, 2021. Clinical data and patient information were obtained from the NTUH-integrative Medical Database of the Department of Medical Research, NTUH. Dataset was structured tabular data with standard fields in the electronic patient file. Steps move from data collection to cleaning included data overview, data dictionary, initial data exploration, consistency check, missing data identification and duplicate detection. Patients with ST-segment elevation myocardial infraction (STEMI) were excluded, as they were promptly referred for emergent revascularization according to guideline-directed therapy; we also excluded cases without available hematological test results or serial hs-cTnT measurement results within 12 hours before diagnosis. The remaining patients diagnosed both with NSTEMI (cases) and non-NSTEMI (eligible controls) were randomly divided into a training and testing set at a ratio of 9:1 after stratification by NSTEMI status for stratified ten-fold cross-validation. ML models incorporated demographic features with 23 NSTEMI-related routine laboratory tests measured from first blood draws to compute patients' risk scores.

For the independent validation set, 10,636 patients with serial hs-cTnT testing results were included from 1 May, 2016, to 31 Dec, 2021 from the Hsin-Chu Branch of NTUH which is a medical center located in central Taiwan with different patient population and has independent healthcare system from the main hospital located in the north. For validation of the model's clinical applicability and its comparison to the 0 h/1 h algorithm, 380 patients with symptoms of myocardial ischemia were identified from the independent external validation cohort. All patients with available hs-cTnT measurements (a time between samples of >0.5 h to ≤1.5 h) were included to evaluate whether combining the ML model with the 0/1 h algorithm could enhance diagnostic accuracy. The confirmation/exclusion criteria of the 0 h/1 h algorithm were based on the 2020 ESC guideline [9] and previous research [19].

### Routine laboratory testing

The routine tests performed in the ED, whether in the testing set or the external validation set, were ordered based on the physician's clinical assessment of the patient's needs. To ensure consistency and reliability of the input features, and to

mitigate the impact of variability in testing methods on model performance, all routine laboratory tests were conducted in independent CAP- and ISO15189-certified medical laboratories for each dataset. If a patient was admitted to the hospital multiple times during the 5-year study period, only the data from the first admission were used, as subsequent visits often represent follow-up or unrelated comorbid conditions. This ensured that the model targeted the initial emergency presentation of chest pain and reduced confounding from post-event or chronic processes. In addition, limiting each patient to a single encounter prevented over-weighting individuals with recurrent visits during model training. Clinical biochemical and hematological tests were carried out using Beckman Coulter AU680 and Sysmex XN-10, respectively. The hs-cTnT analysis was performed using a Roche Cobas e411 analyzer. The units and full names of laboratory tests are listed in S1 Table.

## Feature selection and data preprocessing

Feature selection was performed using data exclusively from NTUH to avoid potential data leakage, without incorporating data from the external validation cohort, and the resulting features were subsequently applied to model training and validation. To retain laboratory tests commonly measured with chest pain and thus make the model parameters more feasible in a real-world setting, we analyzed the models' performance with different missing rate thresholds above which features were excluded. The missing rate analysis showed an obvious gap between BMI and NT-pro BNP (S1A Fig), thus thresholds between 10%-50% excluded the same number of features and resulted in similar model performance (S1B Fig). We excluded tests with >10% missing values according to the data structure. To retain features directly related to AMI status, p-values were calculated by chi-square test after quantile binning, and test results with p-values >0.05 were excluded (S1C Fig). Serum levels of CK, CK-MB, PT, and PT INR (CKs/PTs) were significantly different between cases and controls even though their missing values accounted for >10% of all patients. Therefore, CKs/PTs were retained in the feature vector as it may enhance model performance. Missing values of the dataset were imputed by taking the median value of the available non-missing value from the training set after stratification by sex. This sex-stratified approach reflected established biological differences between males and females in laboratory profiles, enhancing model calibration and reducing bias [21]. Following one-hot encoding, data were normalized to scale each feature to a range between 0 and 1. Patients diagnosed with AMI received a label of "1", while individuals having never been diagnosed with AMI throughout the entire period (5-year time window) were labeled "0".

## AMI diagnosis procedure

The process for diagnosing AMI in the emergency department began with the initial presentation and assessment. Patients presenting with chest pain underwent a rapid evaluation that included a brief inquiry into their symptoms and medical history. A 12-lead ECG was performed within 10 minutes of triage to identify ischemic changes indicative of AMI. For patients exhibiting ST-segment elevation, immediate referral for emergent coronary angiography and revascularization was initiated, following guideline-directed medical therapy. In cases where the ECG did not reveal ST-segment elevation, further evaluation was guided by the clinical presentation and patient history. These patients were admitted for observation with serial diagnostic evaluations, including repeated ECGs and biomarker analysis. Cardiac biomarker assessment involved serial measurements of cTnT. An initial troponin level exceeding the 99th percentile upper reference limit was considered indicative of myocardial injury. If the initial troponin was below the threshold, a repeat measurement was obtained at three hours. A troponin increase exceeding the 99th percentile or a rise of more than 20% compared to the baseline value was diagnostic of NSTEMI.

## Hyperparameter tuning and model construction

Hyperparameters were tuned by using Bayesian optimization [18] with cross-validation on the training set (S2 Table). Models were fitted to the training data containing 27-dimensional feature vectors and a vector of their corresponding labels.

All data processing, ML procedure, and visualization were performed with Python (3.8). Models including random forest (RF) and logistic regression (LR) were constructed using "Scikit-Learn" packages (1.1.2). The XGBoost (XGB) model was constructed using the "xgboost" package (1.5.1). To avoid sensitivity and specificity bias, we evaluated the average values of the area under the receiver-operating-characteristic curve (AUROC) and average precisions of the precision-recall curve (APPRC) performance metrics when performing hyperparameter tuning. The data imbalance problem was addressed by tuning the hyperparameter of class weight or sample weight which is 'scale_pos_weight' for the XGB model and 'class_weight' for the RF as well as LR models. To avoid overfitting, for the XGB model, L1 and L2 regularization was performed by tuning the hyperparameter 'alpha' and 'lambda'; the RF model was regularized by tuning the parameters 'max_depth', 'min_samples_leaf', and 'max_features'; the LR model was regularized by tuning the hyperparameters 'penalty' and 'C'.

### Statistical analysis and model evaluation

Model performances were evaluated by estimating the median and 95% confidence interval (95% CI) of the area under the AUROC and the APPRC from ten-fold cross-validation [17]. For independent validation set, we bootstrapped confidence interval for performance metrics, which is a common strategy for presenting uncertainty [18,19]. AMI risk scores were described by probabilities predicted from ML models. Predefined sensitivity and NPV statistical thresholds determined low-risk thresholds, dividing patients into the low-risk and not-low-risk groups. High-risk thresholds defined by pre-specified specificity and PPV statistical cutoffs divided patients into high- and not-high-risk groups [19,20]. Feature importance estimation for model interpretability was performed using the Shapley Additive Explanations (SHAP) package [22]. The AUROCs were compared using the nonparametric Delong method.

## Results

### Baseline characteristics and NSTEMI machine learning model construction

We firstly designed the study to address an unmet clinical need for patients who cannot be definitively ruled in/out for AMI based on their initial (0 h) hs-cTnT values. Accordingly, we focused on individuals whose diagnosis remained uncertain after initial assessment using symptoms, ECG findings, initial hs-cTnT values, and relevant clinical variables such as medical history, medication use, or other cardiovascular diagnostic information, and who therefore underwent continuous monitoring with serial hs-cTnT evaluations before diagnosis (Fig 1A). Following the exclusion of 157 patients diagnosed with STEMI, 1,256 without CBC data [18], 730 without available hs-cTnT within 12 hours before diagnosis, and 26,761 whose diagnosis was already determined by a single 0 h hs-cTnT measurement and thus not diagnostically challenging or within the primary scope of this study, we established ML models with data from 15,096 patients. Among them, 690 were diagnosed with NSTEMI and 14,406 were eligible controls, patients not diagnosed with AMI, including neither STEMI nor NSTEMI cases (Table 1). The ML model development procedures included data collection, data cleaning, model construction, model performance analysis, and clinical application (Fig 1B). Aiming at rapid diagnosis, we collected routine laboratory data generated along with the first hs-cTnT measurement instead of the second one. A total of 45 laboratory tests, including the first hs-cTnT, were carried out upon admission to the ED (S3 Table). Following feature selection, we developed ML-based models by incorporating demographic features, including age, sex, BMI, and smoking status, with these final 23 tests to generate a 27-dimensional feature vector to predict AMI status.

### Model performance analysis

The XGB model performed best with an AUROC of 0.921 (95% CI: 0.907–0.936) and an APPRC of 0.641 (95% CI: 0.577–0.704) among models, whereas 0 h hs-cTnT alone showed the lowest AUROC of 0.776 (95% CI: 0.751–0.800)

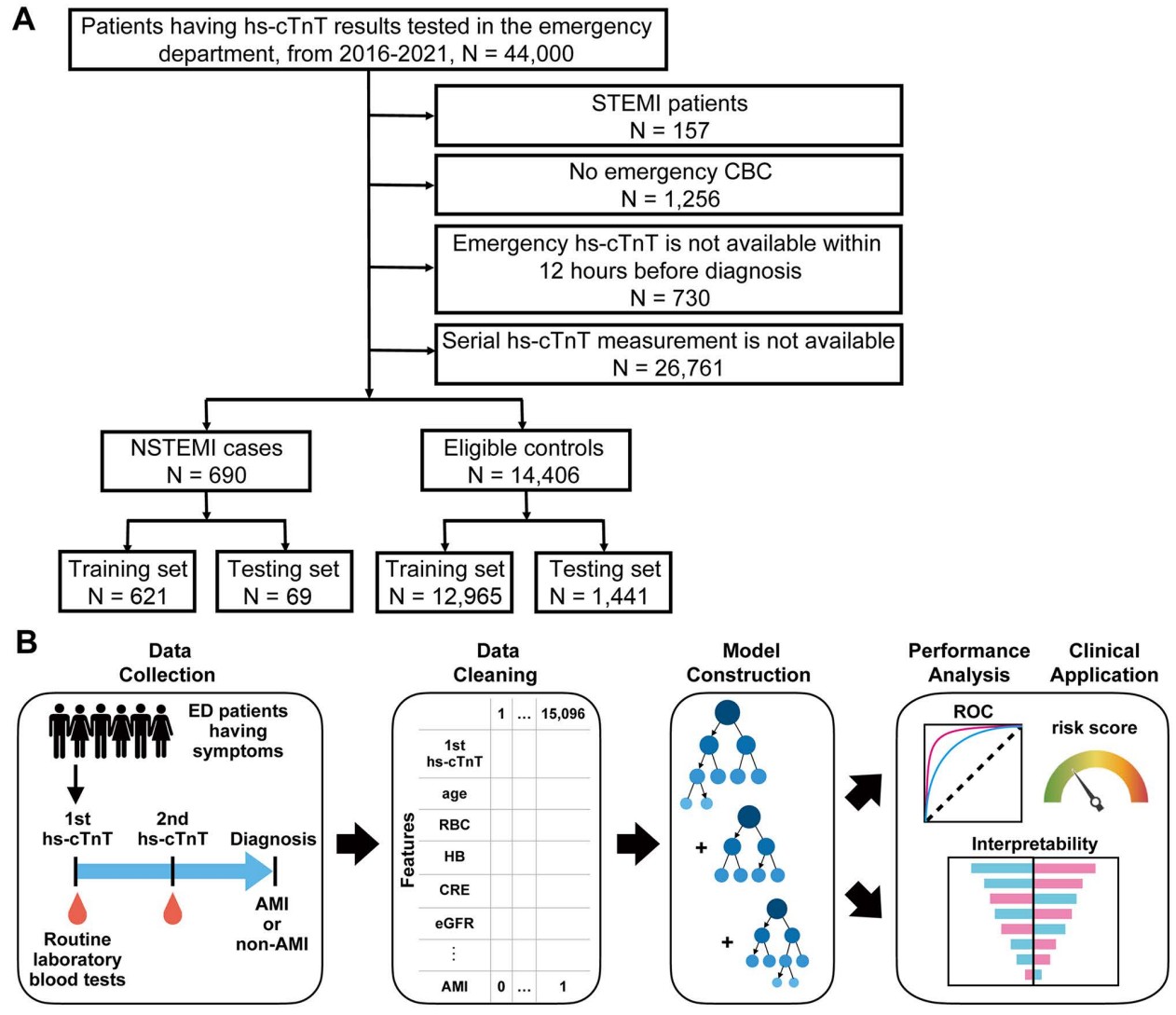

**Fig 1. Patient selection criteria and machine learning procedure. (A)** Inclusion and exclusion criteria cascade of patients in the dataset. STEMI, ST-segment elevation myocardial infarction; NSTEMI, non-ST-elevation myocardial infarction; hs-cTnT, high-sensitivity cardiac troponin T. CBC, complete blood count. **(B)** Schematic diagram of the machine learning model construction pipeline. Routine laboratory testing results completed along with the first hs-cTnT were used to construct a feature vector into which demographics including sex, age, BMI, and smoking status were incorporated. At the endpoint, patients diagnosed with AMI are labeled as "1", while non-AMI status are labeled as "0". ML models were built with the feature vector where each dimension represents a specific laboratory test. Model performances were evaluated by receiver-operating-curve (ROC) and importance analysis.

and APPRC of 0.242 (95% CI: 0.216–0.268) (Fig 2A). We also tested models trained without CKs/PTs, since their missing values >10% despite apparently constituting important features (S1 Fig). The consequent decrease in model performance suggested that CKs/PTs might represent important blood tests that show a potential impact on AMI prediction (S4 Table). To understand each blood test's contribution to the outcome and how models arrived at those predictions, we explored the importance of each feature to the final prediction by the SHAP algorithm. Features displaying high test values with a negative SHAP value tended to negatively affect the output and vice versa. Indeed, hs-cTnT, CK-MB, and CK, serum biomarkers released by necrotic cardiomyocytes, positively affected the output when their values were high (Fig 2B). In contrast, PT INR, PT, and RDW-CV were important features that negatively contributed to AMI prediction because they

**Table 1. Baseline characteristics of the analysis population. Age, sex, BMI, and smoking data are mean (SD) or n (%).**

| | National Taiwan University Hospital | | | | Hsin-Chu Branch of National Taiwan University Hospital | | | |
|---|---|---|---|---|---|---|---|---|
| | Total (n = 15,096) | Case Patients (n = 690) | Control Subjects (n = 14,406) | P-value | Total (n = 10,636) | Case Patients (n = 699) | Control Subjects (n = 9,937) | P-value |
| Age, years (SD) | 69.94 (15.66) | 67.38 (13.50) | 70.07 (15.74) | 1.10E-05 | 69.79 (16.18) | 67.77 (14.09) | 69.94 (16.31) | 6.35E-04 |
| 18-39 years, n (%) | 706 (4.68) | 21 (3.04) | 685 (4.75) | | 566 (5.32) | 17 (2.43) | 549 (5.52) | |
| 40-59 years, n (%) | 2,916 (19.32) | 180 (26.09) | 2,736 (18.99) | | 2,187 (20.56) | 188 (26.90) | 1,999 (20.12) | |
| >60 years, n (%) | 11,474 (76.01) | 489 (70.87) | 10,985 (76.25) | | 7,883 (74.12) | 494 (70.67) | 7,389 (74.36) | |
| Sex | | | | 3.97E-14 | | | | 4.14E-09 |
| Male, n (%) | 8,722 (57.78) | 495 (71.74) | 8,227 (57.11) | | 6,231 (58.58) | 484 (69.24) | 5,747 (57.83) | |
| Female, n (%) | 6,374 (42.22) | 195 (28.26) | 6,179 (42.89) | | 4,405 (41.42) | 215 (30.76) | 4,190 (42.17) | |
| BMI, kg/m$^2$ (SD) | 23.90 (4.13) | 24.69 (3.60) | 23.87 (4.15) | 2.49E-06 | 24.41 (4.36) | 24.96 (4.02) | 24.37 (4.38) | 8.93E-03 |
| Smoking | | | | 2.06E-11 | | | | 9.87E-04 |
| Never, n (%) | 3,838 (25.42) | 109 (15.80) | 3,729 (25.89) | | 2,784 (26.18) | 140 (20.03) | 2,644 (26.61) | |
| Former, n (%) | 402 (2.66) | 4 (0.58) | 398 (2.76) | | 225 (2.12) | 12 (1.72) | 213 (2.14) | |
| Current, n (%) | 410 (2.72) | 18 (2.61) | 392 (2.72) | | 320 (3.01) | 20 (2.86) | 300 (3.02) | |
| Unknown, n (%) | 10,446 (69.20) | 559 (81.01) | 9,887 (68.63) | | 7,307 (68.70) | 527 (75.39) | 6,780 (68.23) | |

had mostly high values when SHAP values were negative. Other important features, such as age, eosinophils, basophils, and lymphocytes, also contributed positively to AMI discrimination.

The results demonstrate the discrimination ability of models trained with routine laboratory tests as features for AMI, while understanding the probable risk and giving emergent medical care according to risk levels are also important in clinical practice. To this end, we adopted the probability generated from the models as a risk score ranging from 0 to 100 for each patient. Higher risk score thresholds gave rise to decreasing patterns of sensitivity and NPV, while lower thresholds resulted in increasing trends of specificity and PPV (S2 Fig). To achieve an NPV ≥ 99.5%, a low threshold of 1.8 (95% CI: 1.2–2.4) was used to divide testing sets into low- and not-low-risk groups, where 61.7% (95% CI: 46.7–76.8%) patients were excluded from the diagnosis (S5 Table). To reach a PPV ≥ 75.0%, a high threshold of 38.5 (95% CI: 32.5–44.4) was used to divide the testing set into high- and not-high-risk groups, where 3.0% (95% CI: 2.4–3.6%) patients were ruled in. These thresholds were selected to align with clinical accepted safety and utility benchmarks for early rule-out and rule-in strategies in ED (NPV ~ 99.0-99.5% for safe discharge and PPV ~ 70.0-80.0% for actionable diagnosis) [19,23].

### Model validation with independent validation set

For independent validation, external data were collected from the geographically distinct Hsin-Chu Branch of NTUH (Table 1). This dataset, comprising patients, principal physicians, healthcare providers, and a healthcare system distinct from those in the training set, served as an appropriate external validation cohort. The XGB model demonstrated a reduced AUROC of 0.864 (95% CI: 0.848–0.879) and an APPRC of 0.525 (95% CI: 0.488–0.560). Similarly, 0 h hs-cTnT

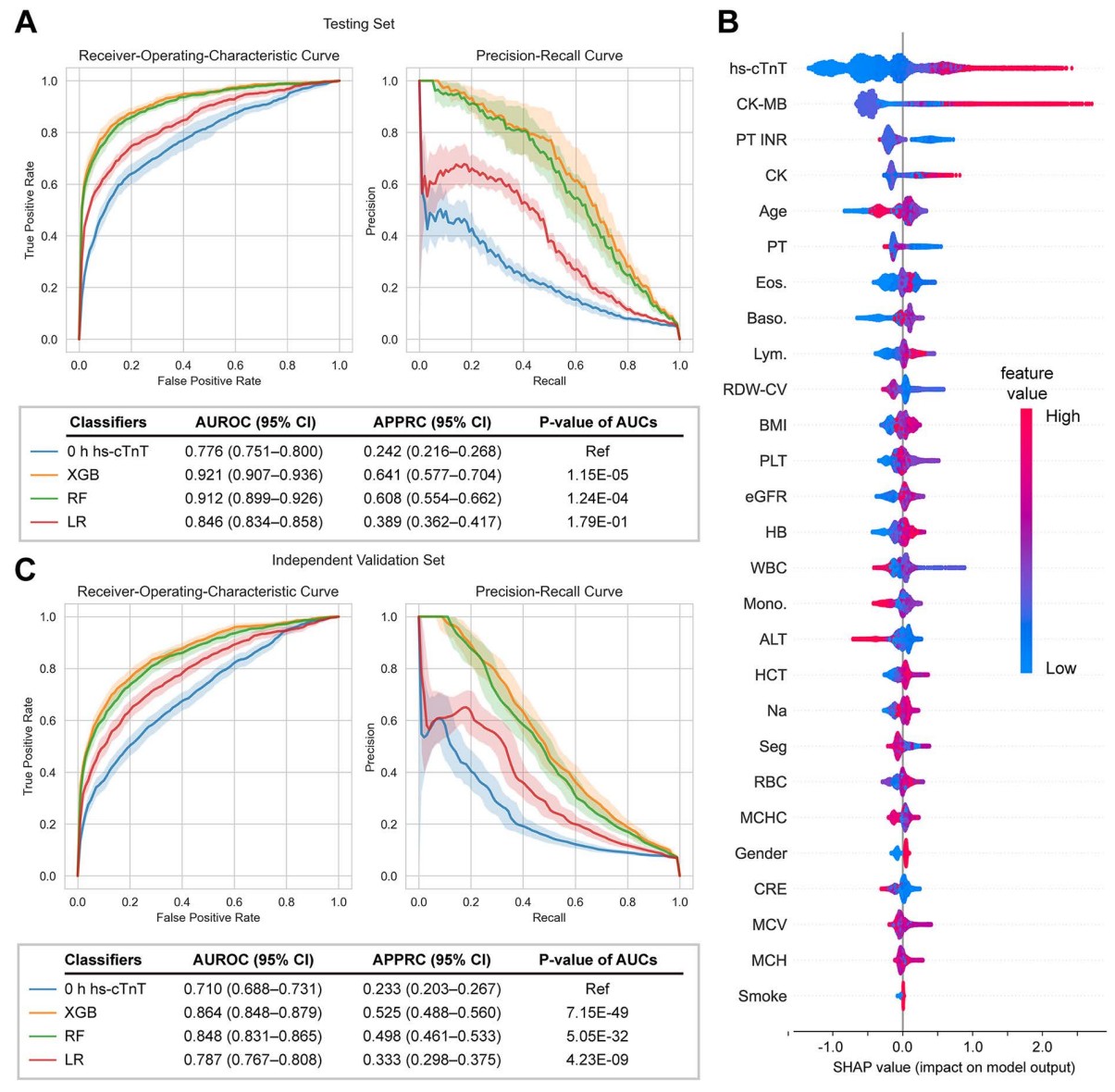

**Fig 2. Prediction performance and interpretability analysis of machine learning models.** Comparison of the area under receiver-operating-curves (AUROC) (upper left) and average precision of precision-recall curves (APPRC) (upper right) generated from the **(A)** testing set and **(C)** independent validation set for the XGBoost (XGB), random forest (RF), and logistic regression (LR) models, and the model using 0 h hs-cTnT alone as an input feature. Solid lines represent median and color zones represent 95% confidence interval. **(B)** Importance analysis explaining the contribution of laboratory tests to predicting the model for AMI status. Feature importance is computed by Shapley Additive Explanations (SHAP) values. The laboratory tests are arranged along the y-axis based on their mean absolute SHAP values depicted on the x-axis. These values signify the influence of the test results on the XGBoost model predictions. The color of individual test values for each patient corresponds to their respective relative values.

exhibited a decreased AUROC of 0.710 (95% CI: 0.688–0.731) and an APPRC of 0.233 (95% CI: 0.203–0.267) (Fig 2C). While this performance decline may be attributed to data distribution shifts, a phenomenon frequently observed when pre-trained models are applied to external datasets, the model still demonstrated superior effectiveness. Given that one of the strengths of the ML algorithm is its ability to provide risk stratification at 0 hours, we evaluated whether these models could enhance predictive power for patients primarily diagnosed based on the first hs-cTnT measurement (n = 26,761,

Fig 1A). Nevertheless, when the algorithm was further applied to these excluded patients, it maintained superior predictive performance compared with the 0 h hs-cTnT alone (S3 Fig), demonstrating that the model possesses good robustness and generalizability to the broader population presenting to the ED, beyond the diagnostically challenging subset originally targeted.

We used the low- and high-risk thresholds derived to meet predefined diagnostic criteria from the testing set (S5 Table) and validated its diagnostic performance for rule-in/out on the independent validation set (Table 2). The risk score threshold of 1.8 resulted in up to 48.3% (95% CI: 47.5–49.3%) exclusion with an NPV of 98.8% (95% CI: 98.5–99.1%). The risk score cutoff of 38.5 ruled in 2.6% (95% CI: 2.3–2.9%) of patients with a PPV of 78.1% (95% CI: 73.2–82.4%). Given that a total of more than 50% of patients could be precisely confirmed or excluded from diagnosis with respective risk score thresholds of 1.8 and 38.5, this algorithm can thus be applied to facilitate diagnosis for patients presenting with suspected NSTEMI directly after the acquisition of their first routine laboratory test results.

## Comparison of the ML model with the ESC 0 h/1 h algorithm for NSTEMI rule-in/out

In clinical practice, the ESC guideline recommends the use of the 0 h/1 h algorithm for rapid triage for patients presenting with suspected NSTE-ACS to the ED. However, approximately one-third of the patients are neither ruled in nor ruled out using this test alone, prompting us to hypothesize that its accuracy might be improved when combined with an ML model. Totally 380 patients with symptoms of myocardial ischemia from the independent external validation cohort and with a time between serial samples of >0.5 h to ≤1.5 h were included to evaluate performances of the 0 h/1 h algorithm (Fig 3A). During the 0 h checkpoint, 63 patients were ruled in with 0 h hs-cTnT measurements with a PPV of 69.8% (95% CI: 61.9–76.2%) and 66 patients were ruled out with an NPV of 98.5% (95% CI: 97.0–100%). When we concatenated the ML and 0 h/1 h algorithm, 66 patients were ruled in with a PPV of 83.3% (95% CI: 77.3–87.9%) and 194 patients were ruled out with an NPV of 100% (95% CI: 99.5–100%) before 1 h hs-cTnT was measured (Fig 3B). This indicates that the incorporation of our algorithm increased ruled-out with improved NPV and elevated the rule-in performance of the 0 h/1 h algorithm with increased PPV during the 0 h checkpoint. After the measurement of 1 h and Δ0–1 h hs-cTnT, 76 [20.0%] patients were ruled in with a PPV of 72.4% (95% CI: 65.8–77.6%), and 185 [48.7%] were ruled out with an NPV of 99.5% (95% CI: 98.9–100%), while 119 [31.3%] were ruled neither in nor out (Fig 3A). When concatenated with the ML algorithm, the 0 h/1 h algorithm showed an improved PPV of 84.9% (95% CI: 79.5–87.7%) for ruling in 73 [19.2%] patients and an NPV of 100% (95% CI: 99.6–100%) for ruling out 251 [66.1%] patients, while only 56 [14.7%] patients were ruled neither in nor out (Fig 3B) during the 1 h checkpoint. The combined results suggest that our ML algorithm not only provided a more rapid approach for AMI diagnosis than the 0 h/1 h algorithm but also reduced the number of patients not qualified for triage by the 0 h/1 h algorithm, without compromising its performance.

Table 2. Model performance on the independent validation set based on different risk score thresholds. Model performance was assessed on the independent validation set based on different risk score cutoffs derived from the testing set to meet predefined criteria for sensitivity, specificity, NPV, and PPV. Low risk score thresholds categorize the population into low-risk and non-low-risk groups, without establishing a high-risk group. Similarly, high risk score thresholds categorize the population into high-risk and non-high-risk groups, without establishing a low-risk group. Data are median (95% CI). NPV, negative predictive value; PPV, positive predictive value.

| Risk score thresholds | Sensitive, % | NPV, % | Specificity, % | PPV, % | Proportion Low Risk, % | Proportion High Risk, % |
|---|---|---|---|---|---|---|
| 1.2 | 95.6 (93.9–97.1) | 99.1 (98.8–99.4) | 34.5 (33.6–35.4) | 9.3 (8.7–10.0) | 32.5 (31.7–33.4) | … |
| 1.8 | 91.3 (89.1–93.3) | 98.8 (98.5–99.1) | 51.1 (50.2–52.1) | 11.6 (10.8–12.5) | 48.3 (47.5–49.3) | … |
| 33.4 | 34.0 (30.7–37.3) | 95.5 (95.1–95.9) | 99.2 (99.0–99.3) | 74.3 (69.6–78.5) | … | 3.0 (2.7–3.4) |
| 38.5 | 30.3 (27.0–33.7) | 95.3 (94.9–95.7) | 99.4 (99.2–99.5) | 78.1 (73.2–82.4) | … | 2.6 (2.3–2.9) |

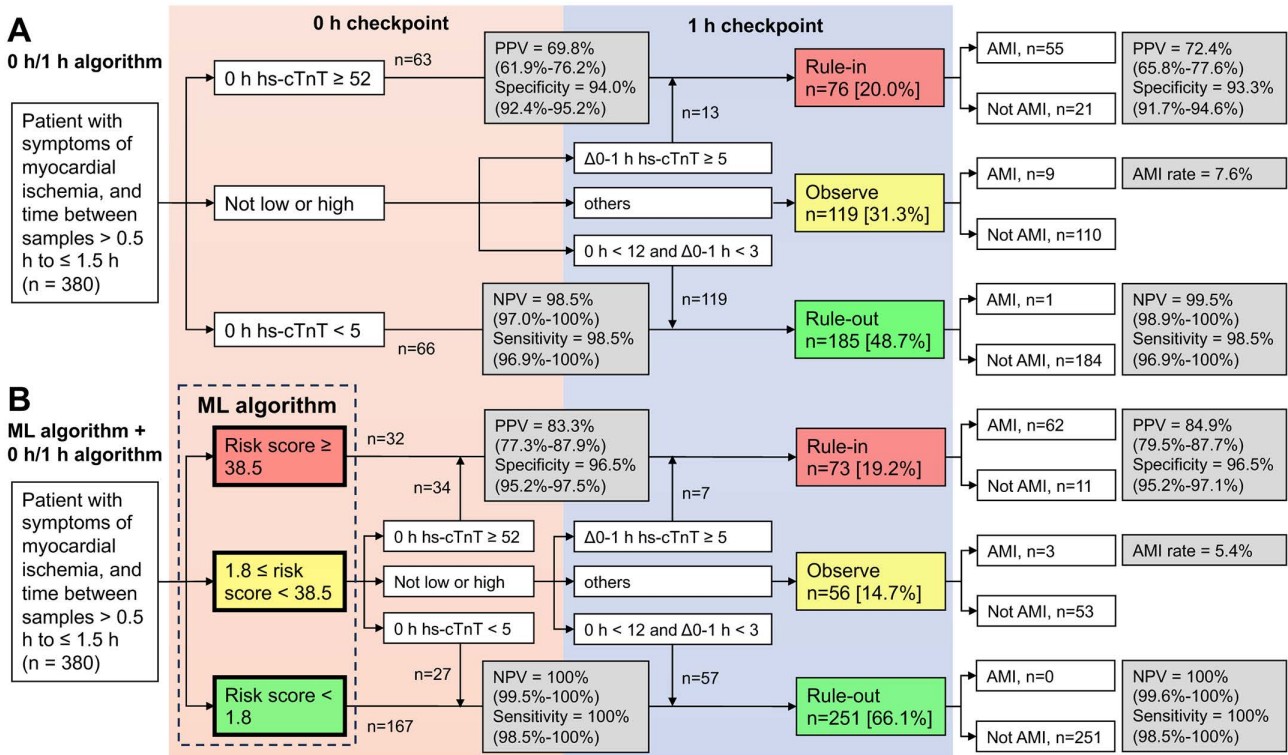

**Fig 3. Comparison of the discriminative capabilities of the machine learning algorithm and the 0 h/1 h algorithm.** Classification cascade of patients with suspected non-ST elevation myocardial infarction using the **(A)** 0 h/1 h algorithm alone or **(B)** in combination with a machine learning algorithm. PPV, positive predictive value; NPV, negative predictive value.

## Discussion

Traditionally, the 0 h/3 h algorithm serves as a criterion for either confirming or ruling out AMI, but its implementation in busy EDs presents challenges and may not be universally applicable. We developed an algorithm that enables swift evaluation of AMI risk using patients' initial routine laboratory test results. Although numerous statistical and neural network models have been proposed for diagnosing and prognosing acute cardiovascular conditions, including AMI [24–26], their clinical adoption has been limited by model complexity, substantial number of required variables insufficient validation [27]. In parallel, a neural network model using a multilayer perceptron (MLP) architecture was also evaluated and achieved an AUROC of 0.877 (95% CI: 0.867–0.888) and an APPRC of 0.511 (95% CI: 0.481–0.540) in the testing set (S4 Fig). While deep learning architectures can capture higher-order nonlinearities, prior evidence suggests that for structured laboratory datasets, gradient boosting methods such as XGB can provide comparable performance while offering greater transparency, interpretability, robustness, and computational efficiency, features essential for clinical adoption in emergency medicine [28]. Therefore, the selection of XGB as the primary modeling approach in this study was guided not solely by predictive performance, but by a balanced consideration of diagnostic accuracy, interpretability, and real-word clinical feasibility to support practical deployment. To address these practical challenges, we validated our models using a large, independent external population and demonstrated improved diagnostic performance across both internal and external cohorts. However, the AUROC decreased from 0.921 to 0.864 with external dataset, highlighting a potential limitation in generalizability. This performance degradation is likely due to data distribution shifts, a phenomenon commonly encountered when pre-trained models are deployed on external datasets [29]. Addressing this issue may involve retraining or

fine-tuning models using site-specific data, especially in settings with limited datasets, to better align the model to local data characteristics and enhance robustness and applicability [30].

Aiming to reduce model complexity and improve discrimination, we selected only important and routinely requested tests by choosing statistically significant features with low missing rates. We used routine hematological and biochemical blood tests including CBC, WBC counts, kidney, liver functions, and electrolytes because their high availability and short turnaround time make them practical for AMI prediction. Our combined results suggest a high diagnostic value of commonly requested blood tests for AMI, while some less frequently ordered tests such as CKs/PTs, despite moderate missingness, still significantly enhanced predictive power. Notably, although CK-MB and CK was not clinically recommended as primary AMI biomarkers due to lower sensitivity and specificity compared to hs-cTnT [7], our SHAP results demonstrated their diagnostic value for NSTEMI, particularly when hs-cTnT results are inconclusive. Prospective studies are needed to further evaluate the clinical and economic impact of including these markers.

We used SHAP analysis, originally a game theoretical approach and is now widely used in medical fields for interpretability [17,18], to identify positive predictors including hs-cTnT, CK-MB, CK, age, basophil, eosinophil, and lymphocyte counts. Although these indices have been individually reported as potential prognostic markers, their combined diagnostic value for AMI is less established. Previous studies typically applied a prespecified cTn threshold for all patients regardless of characteristics like age, sex, or comorbidities that are all known to affect troponin concentrations [31,32]. To capture potential interactions among key variables, we translated hs-cTnT, age, sex, BMI, and smoking status into high-dimensional vector space to determine feature crosstalk. Among the identified predictors, the lymphocyte count, one of the key features displaying high SHAP values, is inversely associated with inflammation, and low lymphocyte count is a poorer prognostic marker in patients with AMI [33]. Conversely, several laboratory parameters, including PT INR, PT, and RDW-CV, acquired higher SHAP values when their test values were low. RDW improves the predictive value for all-cause mortality when combined with the GRACE score and is a potential marker for risk stratification of ACS patients [34]. In a large cohort retrospective study, PT INR, a normalized ratio of PT, was associated with long-term mortality in patients with coronary artery disease [35]. Interestingly, in our cohort, PT levels were lower in AMI cases, despite previous studies reporting increased PT in such patients [36]. This inverse association observed in our study may reflect clinical confounding. For instance, atrial fibrillation (AF) and heart failure (HF) are important confounding factors in NSTEMI diagnosis, as both share overlapping symptoms and affect cardiac biomarkers. Patients with AF or HF often receive anticoagulant therapy, resulting in elevated PT/INR levels. This may explain the negative SHAP contribution of PT/INR, since higher values could reflect anticoagulation or non-ischemic cardiac stress rather acute type 1 MI [37–39]. This discrepancy may result from different study populations, as our study focused on patients who needed second hs-cTnT.

ML models detect subtle changes and allow for flexible thresholds in different conditions, which improves the predictive performance of serial cTn measurements when age and sex are considered [19]. In addition to demographic features, we incorporated routine laboratory tests that were previously shown to be suitable features for ML to predict acute or chronic diseases [17,18]. Although a second hs-cTnT measurement provides greater diagnostic accuracy, its high time demand can be a barrier in clinical settings. Our results show blood tests can be combined to constitute a more powerful index for AMI diagnosis, and the ML algorithm provides risk scores that clinicians can integrate with other clinical information. To explore concrete implementation consequences of the algorithm, we applied risk score thresholds derived to meet pre-specified criteria for sensitivity, specificity, NPV, and PPV [19,20]. The result indicated that about half of patients fell into an intermediate-risk group and may still need subsequent 0 h/1 h algorithms for diagnosis. Nevertheless, compared with the 0 h/1 h algorithm alone in sole use, integrating ML model enabled a greater portion of patients to be definitively triage with improved predictive values. Using only the 0 h sample, our model achieved an NPV of 98.8%, which increased to 100% when integrated with the ESC 0/1 h protocol (Fig 3B), underscoring its clinical utility in safely ruling out NSTEMI and facilitating timely discharge or observation of low-risk patients. This approach may shorten ED stays and improve overall efficiency for both clinicians and patients (Fig 4).

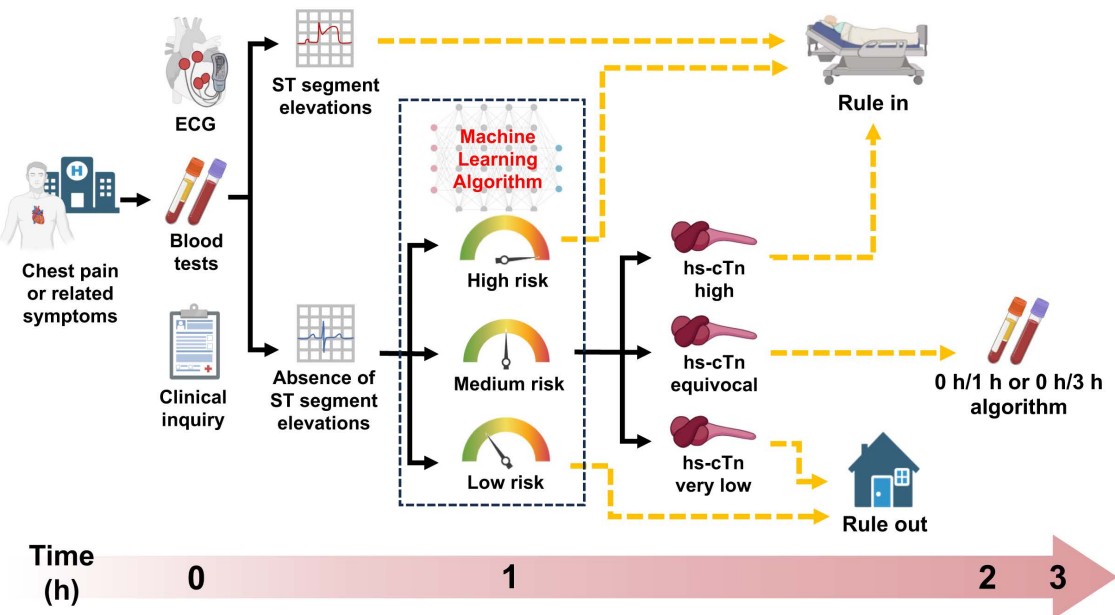

**Fig 4. Visual summary delineates the diagnostic cascade incorporating the machine learning algorithm.** The combined results demonstrate clinically applicable approaches to rapid risk assessment with this machine learning algorithm for patients presenting with suspected non-ST-elevation myocardial infarction to the emergency department. hs-cTn, high-sensitivity cardiac troponin.

Recent studies have demonstrated that deep learning applied to ECG waveform analysis can markedly enhance AMI diagnosis in emergency care [40,41]. While such models effectively capture complex temporal and spatial ECG patterns, our approach complements them by focusing on objective biochemical data that are universally available at presentation. We envision that combining our laboratory-based algorithm with ECG-based deep learning systems could further improve diagnostic accuracy and patient triage in future implementations. Similarly, the MI[3] model proposed by Than et al. employed age, sex, and paired hs-cTnI values through a gradient boosting framework to predict AMI, achieving an NPV of 99.7% and a PPV of 78.1 [19]. Our results were comparable or slightly superior in PPV and specificity, while requiring only a single 0 h blood draw. Moreover, by employing XGB, an advanced and regularized form of gradient boosting, our model achieved enhanced robustness and interpretability. Importantly, rather than serving as a methodological replacement, our algorithm was designed to complement established strategies such as the ESC 0/1 pathway or existing AI tools, thereby supporting a more efficient and clinically adaptable diagnostic workflow.

Several limitations of this study should be acknowledged. First, the observed low prevalence of NSTEMI (~4.5%) is consistent with real-world data [42,43], but this value may have slightly overestimated the NPV and AUC. Recalibration of the model may be necessary when applying it to populations with significantly different disease prevalence. Second, the retrospective nature of this study means that laboratory test ordering in the ED depended on individual physician judgment. This practice-dependent variability led to high proportion of missing values for certain parameters, which might have inadvertently influenced the model's predictive performance. Future multicenter prospective studies are therefore warranted to confirmed the model's generalizability and robustness. Third, while we employed a transparent and practical sex-stratified median imputation approach, we acknowledge that this method may not fully account for complex variable interactions. Adopting more sophisticated techniques, such as multiple or model-based imputation, in future studies could further improve data integrity and model performance. Fourth, the model did not distinguish between Type 1 and Type 2 myocardial infarction, which are clinically relevant due to their distinct management strategies. Accurate subtype

classification often requires additional clinical and imaging information not immediately available upon presentation [44]. Our model focused on early rule-in/rule-out decisions in the emergency setting; subtype-specific evaluation will be the focus of future research using adjudicated cohorts. Finally, we did not assess the prognostic implications of our model, such as its value for predicting mortality or long-term outcomes. Future prospective implementation studies integrating our algorithm into real-world workflows across diverse settings are needed to validate its fairness, assess prognostic performance, and confirm its clinical impact. Overall, despite these limitations, our model demonstrates robust diagnostic performance and immediate clinical applicability, representing a practical step toward accelerating safe and accurate NSTEMI triage in emergency care.

## Supporting information

**S1 Fig. Feature selection criteria for the selection of important features.** (A) The percentage of missing values for each laboratory blood test. The green line indicates the exclusion cutoff where tests with missing values greater than 10% were excluded. (B) Area under the receiver operating characteristic curve (AUROC) (left panel) and average precision of the precision-recall curve (APPRC) (right panel) of models trained with different missing value thresholds. If a feature had missing value proportion > threshold, then the feature was excluded from the feature vector. Data were mean±SD from cross-validation. (C) Features were subjected to quantile binning and p-values were calculated by chi-square test. Features with p-values > 0.05 (green line) were excluded from the feature vector.
(JPG)

**S2 Fig. Sensitivity, specificity, NPV, and PPV determined with different cutoffs of risk score.** Models were tested and the probabilities generated from the XGBoost model were regarded as risk scores. Sensitivity, specificity, NPV, and PPV were estimated and plotted based on different risk score thresholds above which a patient was classified as positive and vice versa. NPV, negative predictive values; PPV, positive predictive values. Solid lines represent medians and color zones represent 95% CI.
(JPG)

**S3 Fig. Prediction performance of models on patients diagnosed based on the first hs-cTnT measurement.** Comparison of the area under receiver-operating-curves (AUROC) (upper left) and average precision of precision-recall curves (APPRC) (upper right) generated from the patients tested only once for hs-cTnT for the XGBoost (XGB) model, random forest (RF) model, logistic regression (LR) model, and model using 0 h hs-cTnT alone as an input feature. Solid lines represent medians and color zones represent 95% CI.
(JPG)

**S4 Fig. Prediction performance of the neural network model.** Comparison of the area under receiver-operating-curves (AUROC) (upper left) and average precision of precision-recall curves (APPRC) (upper right) generated from the testing set for the XGBoost (XGB) model, random forest (RF) model, and multilayer perceptron (MLP) architecture. Solid lines represent medians, and color zones represent 95% CI.
(JPG)

**S1 Table. Full name of the abbreviation of laboratory tests and units.**
(DOCX)

**S2 Table. The setting of tuned hyper-parameters used to train machine learning models.**
(DOCX)

**S3 Table. Summary of the statistics and P-values of routine laboratory tests measured from the first blood draw.**
(DOCX)

**S4 Table. Performance of models trained with or without CKs/PTs features.**
(DOCX)

**S5 Table. Model performance on the testing set based on different risk score cutoffs derived to meet predefined diagnostic criteria.**
(DOCX)

## Author contributions

**Conceptualization:** Wei-Jia Luo, Yih-Mei Liou, Pan-Chyr Yang, Kang-Yi Su.

**Data curation:** Cheng-Han Hsiao.

**Formal analysis:** Wei-Jia Luo.

**Funding acquisition:** Kang-Yi Su.

**Investigation:** Wei-Jia Luo, Cheng-Han Hsiao.

**Methodology:** Cheng-Han Hsiao.

**Project administration:** Wei-Jia Luo, Yih-Mei Liou.

**Resources:** Yih-Mei Liou, Chi-Sheng Hung, Heng-Yu Pan, Chien-Hua Huang.

**Supervision:** Chi-Sheng Hung, Chien-Hua Huang, Pan-Chyr Yang, Kang-Yi Su.

**Validation:** Yih-Mei Liou, Cheng-Han Hsiao.

**Visualization:** Wei-Jia Luo.

**Writing – original draft:** Wei-Jia Luo.

**Writing – review & editing:** Chi-Sheng Hung, Heng-Yu Pan, Chien-Hua Huang, Pan-Chyr Yang, Kang-Yi Su.

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
