## [Decision Letter · Decision Letter 0]

27 Oct 2025

Response to Reviewers
Revised Manuscript with Track Changes
Manuscript
**Journal Requirements:**

1. Please update your online Competing Interests statement. If you have no competing interests to declare, please state: “The authors have declared that no competing interests exist.”

2. In the online submission form, you indicated that “All data and supporting materials have been provided with the article. The Python codes for data processing and visualization of this study are available from the corresponding author upon reasonable request.”.

a) In a public repository,

b) Within the manuscript itself, or

c) Uploaded as supplementary information.

3. Please provide separate main figure files in .tif or .eps format only and ensure that all files are under our size limit of 10MB. You may leave the embedded figures in the manuscript.

For more information about how to convert your figure files please see our guidelines: https://journals.plos.org/digitalhealth/s/figures

4. We have noticed that you have uploaded Supporting Information files, but you have not included a list of legends. Please add a full list of legends for your Supporting Information files before or after the references list.

5. Some material included in your submission may be copyrighted. According to PLOS’s copyright policy, authors who use figures or other material (e.g., graphics, clipart, maps) from another author or copyright holder must demonstrate or obtain permission to publish this material under the Creative Commons Attribution 4.0 International (CC BY 4.0) License used by PLOS journals. Please closely review the details of PLOS’s copyright requirements here: PLOS Licenses and Copyright. If you need to request permissions from a copyright holder, you may use PLOS's Copyright Content Permission form.

Potential Copyright Issues:

Figure 1: Please confirm whether you drew the images / clip-art within the figure panels by hand. If you did not draw the images, please provide (a) a link to the source of the images or icons and their license / terms of use; or (b) written permission from the copyright holder to publish the images or icons under our CC-BY 4.0 license. Alternatively, you may replace the images with open source alternatives. See these open source resources you may use to replace images / clip-art:

- https://openclipart.org/

**Additional Editor Comments (if provided):**
**Reviewers' Comments:**

**Comments to the Author**

1. Does this manuscript meet PLOS Digital Health’s publication criteria?

Reviewer #1: Yes

Reviewer #2: Yes

Reviewer #3: Yes

2. Has the statistical analysis been performed appropriately and rigorously?

Reviewer #1: Yes

Reviewer #2: Yes

Reviewer #3: Yes

3. Have the authors made all data underlying the findings in their manuscript fully available (please refer to the Data Availability Statement at the start of the manuscript PDF file)?

Reviewer #1: Yes

Reviewer #2: No

Reviewer #3: Yes

4. Is the manuscript presented in an intelligible fashion and written in standard English?

Reviewer #1: Yes

Reviewer #2: Yes

Reviewer #3: Yes

Reviewer #1: This manuscript presents a well-executed retrospective study demonstrating the potential of a ML model to enhance NSTEMI risk stratification and triage of patients with suspected ACS at the Emergency Department (ED). The core strength lies in leveraging readily available demographic and routine laboratory data from the initial blood draw alongside high-sensitivity cardiac troponin T (hs-cTnT), effectively accelerating the diagnostic process compared to relying on serial hs-cTn testing alone. Some aspects should be considered:

The overall NSTEMI prevalence is low in the training/testing set (690 cases out of 15,096, ∼4.5%). This inherently inflates the NPV and the AUC. The authors should discuss the expected performance in clinical settings with potentially higher NSTEMI prevalence.

The achieved NPV of 98.8% is slightly lower than published results for the ESC 0/1h algorithm even so the NSTEMI prevalence was much higher in the studies evaluating the 0/1h algorithm.

While the rule-out capability (NPV 98.8%, excluding 48.3% of patients) is acceptable, the rule-in capability (PPV 78.1%, ruling in only 2.6% of patients) is modest in terms of coverage. The major benefit is thus in ruling out NSTEMI and safely accelerating the discharge or observation path for low-risk patients, rather than definitively ruling in the diagnosis for a large cohort. This should be highlighted in the conclusions.

There is potential bias due to 26,761 patients whose diagnosis did not depend on a second hs-cTnT measurement. This means the model was specifically trained on the population that is diagnostically challenging (i.e., those in the "intermediate risk" grey zone that require serial testing). This limits the model’s generalizability to other population presenting to the ED.

The routine lab tests were ordered based on the physician’s clinical assessment. This could artificially inflate model performance in the retrospective setting, which should be addressed in the limitation setting.

Missing values were imputed using the median value stratified by sex. While acceptable for initial analysis, this simple imputation method may fail to capture the complex distribution of clinical data and could introduce bias, especially since some retained features like CKs/PTs had >10% missing values. The authors adequately tested models without these features, but a more sophisticated imputation method (e.g., Multiple Imputation) should be considered.

In my opinion important confounders of NSTEMI diagnoses are missing in the dataset: especially, heart failure and atrial fibrillation. The letter could explain the negative contribution of PT INR in the SHAP analysis.

Current challenges in the diagnostic triage of suspected MI are not addressed: Especially, the differentiation of type I and type II MI.

Reviewer #2: Summary: Summary

The paper addresses a clinically relevant problem, which is that the timely diagnosis of NSTEMI is challenging, as current protocols rely on serial hs-cTn tests that may delay clinical decisions and overcrowd emergency departments. In this retrospective study, ML models (XGBoost, Random Forest and Logistic Regression) were experimented with using demographics and routine lab tests to develop a predictive model and hence enable early NSTEMI risk stratification. The final ML model consistently outperformed hs-cTn alone in both internal and external validation sets in terms of AUROC. At clinically relevant thresholds, it achieved a negative predictive value (NPV) of 98.8% (98.5–99.1%) for rule-out and a positive predictive value (PPV) of 78.1% (73.2–82.4%) for rule-in, encompassing 48.3% and 2.6% of patients, respectively. When combined with the established 0 h/1 h algorithm, the ML model further enhanced early decision-making, safely ruling in/out 85.3% of patients within 1 hour, with PPV and NPV reaching 84.9% (79.5–87.7%) and 100% (99.6–100%), respectively. These findings highlight the potential of ML-enhanced triage to accelerate diagnosis and reduce unnecessary observation in the emergency department.

Strengths:

1. Clinical Relevance and Innovation: The study focuses on patients who could not be definitively ruled in or out at presentation, highlighting a clinically challenging subgroup and demonstrating the practical utility and innovation of the proposed diagnostic approach.

2. Robust Design and Generalizability: The inclusion of both internal and independent external cohorts enhances the study’s reliability and generalizability. While some fine-tuning may be required for the external data, the consistent performance gains achieved through machine learning integration are encouraging.

3. Enhanced Diagnostic Performance: Combining the machine learning model with the established 0 h/1 h algorithm improved diagnostic accuracy and enabled faster patient triage.

4. Presentation: The study is clearly written, supported by effective figures, with the graphical abstract clearly illustrating the diagnostic process and key outcomes.

Weaknesses:

1. It is unclear on which validation cohorts the combined ML + 0 h/1 h algorithm was evaluated: both internal and external, or only a subset? If it were conducted only on an internal set, extending the validation to an external set can provide a meaningful assessment of the discussed framework.

2. The study lacks a thorough discussion of its limitations. Two areas, in particular, warrant further attention. First, the absence of deep learning approaches is a notable gap, as such models have consistently demonstrated superior capability in capturing complex ECG patterns and may offer enhanced diagnostic performance [1, 2]. Second, the study does not sufficiently address how inter-subject variability in ECG signals was managed, which is especially important in large and diverse population studies [3]. Beyond, inter-subject variabitlity, assessing the model across subgroups (for example, age groups, genders and ethnicities) can help understand the model’s strengths and limitations along with demonstrating the framework’s applicability and fairness. A detailed discussion of these aspects would strengthen the study’s transparency and contextualize its findings within the broader methodological landscape.

[1] Hannun et al., 2019. Cardiologist-level arrhythmia detection and classification in ambulatory electrocardiograms using a deep neural network.

[2] Devkota et al., 2025. AI analysis for ejection fraction estimation from 12-lead ECG.

[3] Gyawali et al., 200. Learning to disentangle inter-subject anatomical variations in electrocardiographic data.

Quetions:

1. What is the reason behind using only the data from the first admission in case a patient was admitted to the hospital multiple times? Is there any significant reason?

2. Why was a random split chosen to split the data into train and test sets? Could there be other criteria?

3. What is the reason for performing missing data imputation based on gender?

4. Thresholds of 1.8 and 38.5 were applied to define low-risk and high-risk groups. Beyond achieving target sensitivity, specificity, PPV, and NPV, do these thresholds have any additional clinical or statistical justification?

Reviewer #3: Overall:

The authors demonstrate the use of machine learning algorithms to classify NSTEMI patients and non-NSTEMI patients with high AUC on two separate datasets. They clearly demonstrated the utility of their algorithm in the clinical workflow for diagnosing acute myocardial infarction and demonstrated significant generalizability of their model. I think this is a strong manuscript; however, I think a few more comparisons with other established methods in the literature would help contextualize this work. My comments are below:

Comments

• It was not immediately clear why you excluded STEMI patients until I read “For patients exhibiting ST-segment elevation, immediate referral for emergent coronary angiography and revascularization was initiated, following guideline-directed medical therapy” in the supplementary material. I would add this sentence to the methods section when introducing the exclusion criteria.

• Were the 14,406 controls people that were not diagnosed with an AMI? This is a bit unclear in the manuscript, please clarify.

• For the feature selection, was the analysis shown in Fig S1 using just the data from the NTUH? Or did it include both datasets? Ideally it was done just on the first dataset to avoid data leakage.

• You mention in the discussion that you wanted to avoid the use of neural networks to maintain simplicity and interpretability. However, it would be interesting to baseline your method against a more complex algorithm to check that you are not sacrificing a significant amount of model performance for interpretability. SHAP analysis can also be used for neural network models to uncover feature importance as well.

• Additionally, it would be useful to compare your method to other methods presented in the literature before you like the MI^3 algorithm in your references ([19, 20]). They use a gradient boosting algorithm which was not tried in this work.

**Do you want your identity to be public for this peer review?** For information about this choice, including consent withdrawal, please see our Privacy Policy

Reviewer #1: **Yes:** Nils Sörensen

Reviewer #2: No

Reviewer #3: No

**Figure resubmission:**

**Reproducibility:** To enhance the reproducibility of your results, we recommend that authors of applicable studies deposit laboratory protocols in protocols.io, where a protocol can be assigned its own identifier (DOI) such that it can be cited independently in the future. Additionally, PLOS ONE offers an option to publish peer-reviewed clinical study protocols. Read more information on sharing protocols at https://plos.org/protocols?utm_medium=editorial-email&utm_source=authorletters&utm_campaign=protocols

---

## [Decision Letter · Decision Letter 1]

8 Feb 2026

First-line risk stratification with machine learning models facilitates rapid triage for non-ST-elevation myocardial infarction

PDIG-D-25-00558R1

Dear Dr. Su,

We are pleased to inform you that your manuscript 'First-line risk stratification with machine learning models facilitates rapid triage for non-ST-elevation myocardial infarction' has been provisionally accepted for publication in PLOS Digital Health.

Best regards,

Bishal Lamichhane

Guest Editor

PLOS Digital Health

**Additional Editor Comments (if provided):**

**Reviewer Comments (if any, and for reference):**

Reviewer's Responses to Questions

**Comments to the Author**

Reviewer #1: All comments have been addressed

Reviewer #2: All comments have been addressed

publication criteria?

Reviewer #1: Yes

Reviewer #2: Yes

3. Has the statistical analysis been performed appropriately and rigorously?

Reviewer #1: Yes

Reviewer #2: Yes

4. Have the authors made all data underlying the findings in their manuscript fully available (please refer to the Data Availability Statement at the start of the manuscript PDF file)?

Reviewer #1: Yes

Reviewer #2: Yes

5. Is the manuscript presented in an intelligible fashion and written in standard English?

Reviewer #1: Yes

Reviewer #2: Yes

Reviewer #1: The manuscript improved substantially. Especially the limitations section now clearly points out remaining weaknesses of the dataset and study design. Overall, I think it is a well written manuscript and should be of interest to the readership of PLOS Digital Health.

Reviewer #2: (No Response)

**Do you want your identity to be public for this peer review?** For information about this choice, including consent withdrawal, please see our Privacy Policy

Reviewer #1: **Yes:** Nils Sörensen

Reviewer #2: No
